# DiSTRICT: Dialogue State Tracking with Retriever Driven In-Context Tuning

**Praveen Venkateswaran, Evelyn Duesterwald, Vatche Isahagian**
IBM Research
{praveen.venkateswaran, duester, vatchei}@ibm.com

## Abstract

Dialogue State Tracking (DST), a key component of task-oriented conversation systems, represents user intentions by determining the values of pre-defined slots in an ongoing dialogue. Existing approaches use hand-crafted templates and additional slot information to fine-tune and prompt large pre-trained language models and elicit slot values from the dialogue context. Significant manual effort and domain knowledge is required to design effective prompts, limiting the generalizability of these approaches to new domains and tasks. In this work, we propose DiSTRICT, a generalizable in-context tuning approach for DST that retrieves highly relevant training examples for a given dialogue to fine-tune the model without any hand-crafted templates. Experiments with the MultiWOZ benchmark datasets show that DiSTRICT outperforms existing approaches in various zero-shot and few-shot settings using a much smaller model, thereby providing an important advantage for real-world deployments that often have limited resource availability.

## 1 Introduction

Task-oriented dialogue systems are increasingly used to enable users to perform tasks through multi-turn conversations in various domains such as travel reservations, banking transactions, or online shopping. Dialogue state tracking (DST) is a critical component of these systems that tracks user requirements by determining key information at each turn in the dialogue (Jacqmin et al., 2022). Given a predefined schema of task parameters (i.e., slots), DST models identify and represent the dialogue state as pairs of slots and their corresponding values as shown in Figure 1.

In real-world deployments, new task domains and slots are frequently added to improve user functionality. Hence, periodic updates to the models may be required, even when the new domains offer little to no dialogue data for training. To address

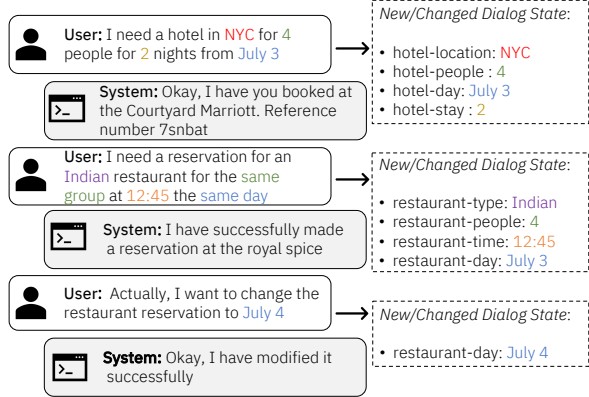

Figure 1: An example of multi-domain dialogue state tracking (DST) in a conversation to book a hotel stay and reserve a table at a restaurant.

these challenges, DST solutions need to be generalizable to new zero-shot and few-shot settings with minimal overhead while also maintaining a small model footprint to enable compute-efficient and cost-effective deployments.

Recent advances in DST have leveraged pretrained language models (LMs) to elicit slot values from dialogues using primarily two methods – fine-tuning and in-context learning. However, both methods suffer from several shortcomings –

**Fine-tuning LMs –** Most existing approaches condition LMs for DST by fine-tuning their parameters using prompts derived from historical dialogue data. However, they typically rely on handcrafted prompt templates that include slot-specific questions, value based functions, or even text-to-code snippets (Lin et al., 2021b; Cao and Zhang, 2021). In addition to the manual effort required, these templates are customized to specific domains and slots, and hence have low generalizability to new domains. Some approaches also extend finetuning to first train models on additional natural language tasks using external datasets which is expensive and requires access to significantly more data. They also often rely on additional information

from the schema such as slot descriptions and task instructions (Mi et al., 2022). However, real-world datasets may not always have this necessary information, which again limits their generalizability.

**In-context learning** – LMs have shown remarkable performance through in-context learning of new tasks (Brown et al., 2020), where a *raw* LM (i.e. pre-trained LM without fine-tuning on task-specific data) is prompted during inference using input-output task examples to condition the generated output. For DST, approaches have leveraged this to craft prompts containing examples of dialogue history and slot values (Hu et al., 2022). However, similar to fine-tuning approaches, these prompt examples are hand-crafted, customized, and require significant effort. Additionally, as a result of using raw LMs, these approaches have to rely on extremely large models limiting their practical use. Importantly, it has been shown that prompting raw LMs without fine-tuning is often oversensitive to example choices and instruction wording (Chen et al., 2022), and can demonstrate undesirable biases that significantly reduce performance (Zhao et al., 2021; Liu et al., 2021).

In this work, we address these challenges and present DiSTRICT, a generalizable approach for dialogue state tracking that fine-tunes a LM using relevant examples (i.e., *in-context tuning*). For a given input dialogue and slot to be tracked, DiSTRICT retrieves semantically similar dialogues and slots from available historical data in zero-shot and few-shot settings, and concatenates them into a prompt with no hand-crafted template requirements. We first fine-tune the language model using in-context examples and input dialogues from the training set, and subsequently perform similar inference on test inputs as shown in Figure 2. Specifically, we make the following contributions –

- To the best of our knowledge, DiSTRICT is the first DST approach to use in-context tuning by fine-tuning a LM with in-context examples.

- DiSTRICT avoids shortcomings of prior approaches by leveraging relevant existing dialogue and slot examples without requiring hand-crafted prompts or external datasets, thereby improving generalizability and avoiding manual overhead.

- Our evaluation shows that DiSTRICT outperforms existing approaches on most zero-shot

and few-shot settings, while using a much smaller model, thus demonstrating its practicality and applicability to real-world deployments.

## 2 Related Work

Dialogue state tracking (DST) is a critical yet challenging task for task-oriented dialogue systems (Williams et al., 2014), and several multi-domain benchmark conversation datasets have been proposed (Budzianowski et al., 2018; Eric et al., 2019; Rastogi et al., 2020) to evaluate research efforts.

A majority of state-of-the-art approaches fine-tune language models using hand-crafted templates containing descriptions or questions related to the dialogue slots. Shah et al. (2019) used slot descriptions and examples of slot values to create templates while Lin et al. (2021b) and Lee et al. (2021) provided different types of manually annotated slot descriptions to the model. Mi et al. (2022) extended this by also including task instructions and other constraints. In contrast, our approach does not require hand-crafted templates for fine-tuning and is hence more easily generalizable.

Another set of approaches aim to improve zero-shot performance by exploiting external knowledge and datasets from other natural language tasks before fine-tuning a model for DST. For instance, Gao et al. (2020); Li et al. (2021); Lin et al. (2021a) pre-train models on reading comprehension data, Shin et al. (2022) reformulate DST as a dialogue summarization task with external annotated data, and Hudeček et al. (2021) use semantic analysis and named entity recognition to identify slots. In contrast, our approach does not require any extra datasets or training efforts .

In-context learning (ICL) for DST has been explored as part of a larger set of few-shot generative tasks (Madotto et al., 2021; Xie et al., 2022), but the lack of a task-specific prompt design resulted in low performance. Hu et al. (2022) and Gupta et al. (2022) achieved improved performance using extremely large models, where the former reformulated DST as a text-to-SQL task by using semantic matching to identify relevant examples that were subsequently crafted into SQL queries, and the latter manually created example dialogues containing combinations of slots in the schema.

Recent efforts have shown that the shortcomings of ICL (Liu et al., 2022; Min et al., 2022) can be overcome through in-context tuning of LMs. Gu-

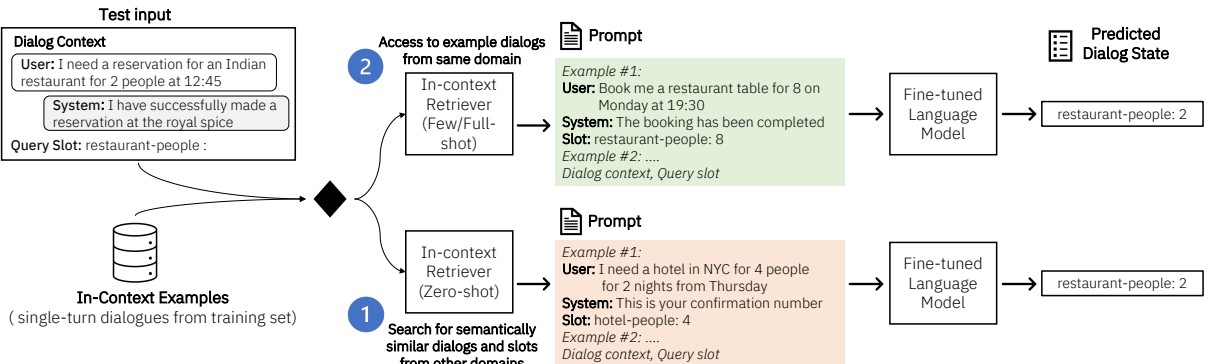

Figure 2: Overview of DiSTRICT. Given an input dialogue and slot to be tracked, high-relevance examples are first identified by the retriever and used to create the prompt. In (1) zero-shot settings, the retriever must search for semantically similar dialogue contexts and slots from other domains, while in (2) few/full-shot settings, the retriever additionally has access to some example dialogues from the same domain that could also contain the query slot.

rurangan et al. (2020) and Liu et al. (2022) demonstrate the improved performance of models fine-tuned with examples compared to ICL over a variety of language tasks. In this work, we leverage this concept specifically for DST.

## 3 Approach

We first present the background and some definitions for dialogue state tracking before describing our approach.

### 3.1 DST Background

A task-oriented dialogue consists of a multi-turn conversation between a user $U$ and the system $A$. Given a dialogue context $C_t$ as the sequence of utterances until turn $t$, (i.e.) $C_t = [A_1, U_1, ..., A_t, U_t]$, the goal of DST is to predict the dialogue state $y_t$, defined as a set of *(slot, value)* pairs:

$$y_t = \{(s_t^i, v_t^i) \mid C_t , \forall s^i \in \mathcal{S}\}$$

where $\mathcal{S}$ denotes the set of possible slots predefined in an ontology or schema. In a multi-domain setting, the schema can comprise of different domains or topics, each corresponding to a service such as restaurant booking or banking. The slots associated with each domain can be either categorical with a set of candidate values (e.g. restaurant-open = 'True' / 'False'), or non-categorical, where the value is a span in the dialogue context (e.g. hotel-name = 'Courtyard Marriott').

### 3.2 In-Context Retriever

The key concept behind our approach is the identification of the most semantically relevant *in-context*

*examples* from the available training set of dialogues. Intuitively, historical labeled dialogues contain information about slots and their values under different conversational contexts. Hence, for an input dialogue and given query slot, conditioning the model during fine-tuning using example dialogues that are semantically similar and additionally contain the same or similar slots, enables the model to better learn the association between slots, their values, and the context.

As shown in Figure 2, the retriever in DiSTRICT performs semantic matching of the input dialogue and slot with single-turn training set conversations as examples (i.e. one pair of user-system utterances). This design choice is due to the fact that large prompts require additional memory and compute, significantly increasing the training time of the model. Furthermore, real-world dialogues can be lengthy, and the context needed to find the value of a particular slot can often be limited to a single sentence. Hence, constraining the in-context examples to single-turn conversations reduces the prompt size, enables the addition of more examples, and removes irrelevant dialogue context.

Formally, we define a dataset $\mathcal{D} = \{(e_j, s_j^i, v_j^i)\}$ consisting of single-turn dialogue examples $e_j$, containing an observed slot $s_j^i$ and its corresponding value $v_j^i$. For a given input dialogue context $\mathcal{C}_t$ and query slot $s^q$, we retrieve the $k$ most relevant examples $\mathcal{E}_k \in \mathcal{D}$ based on the similarity between their text embeddings –

$$\mathcal{E}_k = \max_k \{\text{sim}[(\mathcal{C}_t \oplus s^q)^{\text{emb}} , \mathcal{D}^{\text{emb}}]\}$$

where $\oplus$ denotes concatenation.

### 3.3 Applicability to Zero-shot and Few-shot

To illustrate the generalizability of the retriever to zero-shot and few-shot settings, we use the example shown in Figure 2. Given a test input dialogue from the restaurant domain and the query slot `restaurant-people`, the retriever identifies the $k$ most relevant single-turn in-context examples derived from the training set.

In a zero-shot setting (Figure 2-(1)), dialogue and slot examples from the restaurant domain would not be available. Hence, the retriever identifies semantically similar examples from other domains. For instance, conversations about hotel reservations have similar contexts, and the slot `hotel-people` is semantically similar to the query slot. The retrieved example thus conditions the model to look for the number of people mentioned in the dialogue.

In few-shot and full-shot settings (Figure 2-(2)) the set of available examples would include other dialogues from the same domain which could also contain the query slot. Hence, the most similar examples retrieved would demonstrate the values of the query slot when used in similar restaurant reservation contexts (e.g. `restaurant-people: 8`). We note that our approach requires no changes for the different settings, and can be easily extended to include additional information like slot descriptions to further enhance the semantic retrieval.

### 3.4 In-context Tuning

We fine-tune the language model by retrieving in-context single-turn dialogue examples for each dialogue in the training set. As shown in Figure 2, to create the input to the model, we annotate prefixes to each of the $k$ in-context examples $\mathcal{E}_k$, dialogue context $\mathcal{C}_t$, and the query slot $s^q$ to enable the model to distinguish between them, and then concatenate them into a single input sequence. We then fine-tune an encoder-decoder based language model, where the input is passed to the encoder, and the decoder generates the corresponding value for the query slot. The model in-context tuning objective $\mathcal{L}$ is to minimize the negative log-likelihood loss –

$$\mathcal{L}(\theta) = -\sum_{i=0}^{n} \log p(v_i \mid \mathcal{E}_k \oplus \mathcal{C}_t \oplus s^q)$$

where $n$ is the total number of slots in the ontology and $\oplus$ denotes concatenation.

## 4 Experiments

| Model | #Parameters | Size Diff. |
|---|---|---|
| STARC | 355M | 17× |
| T5DST | 60M | 1× |
| TransferQA | 770M | 13× |
| DS2-BART | 406M | 7× |
| DS2-T5 | 770M | 13× |
| IC-DST | 175B | 3K× |
| DiSTRICT | 60M | |

Table 1: Comparison baselines and their pre-trained model size. TRADE (Wu et al., 2019) does not use a pre-trained model.

### 4.1 Datasets and Evaluation

MultiWOZ (Budzianowski et al., 2018) is a multi-domain task-oriented dialogue benchmark dataset that consists of around 10k multi-turn dialogues over 7 domains. The dataset has been refined and erroneous annotations have been corrected over multiple versions. To enable comparisons with most existing approaches, we use the MultiWOZ 2.0 and MultiWOZ 2.1 (Eric et al., 2019) datasets in our evaluation, and follow the same data pre-processing and domain selection steps as prior work (Wu et al., 2019; Gao et al., 2020; Lin et al., 2021b). To make consistent comparisons to prior work in zero-shot and few-shot settings, we use the same Joint Goal Accuracy (JGA) metric to evaluate our approach. For a given turn, JGA compares predicted dialogue states to the corresponding ground truth states and considers the prediction as accurate if and only if all predicted values match the ground truth values (Wu et al., 2019; Wen et al., 2017; Kumar et al., 2020).

### 4.2 Implementation

DiSTRICT uses T5-small (60M parameters) (Raffel et al., 2020) which is one of the smallest pre-trained models available[1]. It has 6 encoder-decoder layers and the size of each layer is 512. We fine-tune using an AdamW optimizer (Loshchilov and Hutter, 2018) with an initial learning rate of $1e-4$. For the retriever, we use Sentence-BERT (Reimers and Gurevych, 2019) and perform semantic search with cosine similarity using the FAISS library (Johnson et al., 2019). Unless specified otherwise, we set the number of in-context examples to be $k = 3$. We use a single NVIDIA V100 GPU

---

[1]https://huggingface.co/transformers/v2.9.1/pretrained_models.html

| Model | Attraction | Hotel | Restaurant | Taxi | Train | Avg. |
|---|---|---|---|---|---|---|
| | | | MultiWoz 2.0 | | | |
| TRADE | 19.87 | 13.70 | 11.52 | 60.58 | 22.37 | 25.76 |
| T5DST | 33.09±1.60 | 21.21±0.61 | **21.82±0.91** | 65.09±0.12 | 35.42±1.42 | 35.20±0.59 |
| DiSTRICT | **33.61±0.73** | **22.82±0.43** | 21.30±1.08 | **66.53±0.21** | **46.61±1.56** | **38.17±0.80** |
| | | | MultiWoz 2.1 | | | |
| TRADE | 20.06 | 14.20 | 12.59 | 59.21 | 22.39 | 25.69 |
| TransferQA | 31.25 | **22.72** | **26.28** | 61.87 | 36.72 | 35.77 |
| DiSTRICT | **33.74±0.80** | 22.29±0.29 | 24.17±0.94 | **66.34±0.79** | **46.93±1.64** | **38.69±0.89** |

Table 2: Zero-shot joint goal accuracy (JGA) on MultiWOZ 2.0 and 2.1. Results for TRADE and T5DST from Lin et al. (2021b), and TransferQA from Lin et al. (2021a).

for our experiments and provide further details in the appendix.

## 4.3 Comparison Baselines

We evaluate DiSTRICT against existing DST baselines as shown in Table 1. The table shows that, except for T5DST which also uses T5-small, prior DST approaches use models that are significantly larger compared to DiSTRICT.

**TRADE** (Wu et al., 2019) uses slot and domain embeddings as well as a copy mechanism to track slot values across domains.

**STARC** (Gao et al., 2020) prompts two different instances of the RoBERTa-large (Liu et al., 2019) model with separate natural language questions for categorical and non-categorical slots.

**T5-DST** (Lin et al., 2021b) fine-tunes a T5-small model (Raffel et al., 2020) with multiple hand-crafted prompts that include questions and different descriptions of slots.

**TransferQA** (Lin et al., 2021a) represents DST as a QA task, where the model is pre-trained on six external QA datasets and individual questions are manually crafted for each slot in the ontology to use in the prompt.

**DS2** (Shin et al., 2022) treats DST as a dialogue summarization task, and fine-tunes T5-large and BART models with synthetic summary templates.

**IC-DST** (Hu et al., 2022) reformulates DST as a text-to-SQL task and transforms relevant in-context examples to SQL queries and prompts a Codex model without any fine-tuning.

## 4.4 Experimental Settings

**Zero-shot –** Similar to prior work (Lin et al., 2021a; Wu et al., 2019), the retriever and model have access to training data from all domains except from one 'unseen' domain, on which the model is evaluated. We note that our retriever does not result in any information leakage since no examples and slots are included from the unseen domain.

**Cross-domain few-shot –** We include three few-shot settings, where the retriever and model additionally have access to $1\%, 5\%$, and $10\%$ of training data from the unseen domain, similar to Shin et al. (2022); Wu et al. (2019).

**Multi-domain few-shot –** We follow the multi-domain scenario from Shin et al. (2022); Wu et al. (2020), where $1\%, 5\%$, and $10\%$ of the entire training data are sampled for model training and retrieval.

**Note:** We do not include the zero-shot results from prior in-context learning work (Hu et al., 2022) (IC-DST) since their prompt examples are designed to include information from the 'unseen' domain, which results in information leakage to the model, and hence does not reflect the traditional zero-shot learning setting. Additionally, we include results from the other comparison approaches where available.

## 4.5 Results

**Zero-shot DST**

Table 2 shows the dialogue state tracking performance of DiSTRICT in zero-shot settings along with the available baseline results for TRADE, T5DST, and TransferQA. We observe that DiSTRICT outperforms the baseline approaches in most domains across both datasets. DiSTRICT achieves an $8\%$ improvement in JGA on average

| Model | Attraction | | | Hotel | | | Restaurant | | | Taxi | | | Train | | |
|---|---|---|---|---|---|---|---|---|---|---|---|---|---|---|---|
| | 1% | 5% | 10% | 1% | 5% | 10% | 1% | 5% | 10% | 1% | 5% | 10% | 1% | 5% | 10% |
| TRADE[†] | 35.88 | 57.55 | 63.12 | 19.73 | 37.45 | 41.42 | 42.42 | 55.70 | 60.94 | 63.81 | 66.58 | 70.19 | 59.83 | 69.27 | 71.11 |
| STARC* | 40.39 | 65.34 | 66.27 | 45.91 | 52.59 | 57.37 | 51.65 | 60.49 | 64.66 | 72.58 | 75.35 | 79.61 | 65.67 | 74.11 | 75.08 |
| TransferQA* | 52.3 | 63.5 | 68.2 | 43.4 | 52.1 | 55.7 | 51.7 | 60.7 | 62.9 | 75.4 | 79.2 | 80.3 | 70.1 | 75.6 | 79.0 |
| T5DST[†] | 58.77 | 65.72 | 69.54 | 43.07 | 50.71 | 54.86 | 57.63 | 61.86 | 63.47 | 70.12 | 73.67 | 74.70 | 70.82 | 74.18 | 77.57 |
| DS2 - T5* | 65.26 | 69.40 | 70.89 | 44.34 | 52.16 | 53.79 | 58.94 | 64.12 | 64.65 | 74.15 | 77.18 | 78.50 | 74.21 | 76.96 | 78.60 |
| DiSTRICT | **72.14** | **74.61** | **75.91** | **48.07** | **58.84** | **59.29** | **61.17** | **69.31** | **70.07** | **82.65** | **85.37** | **85.89** | **79.77** | **81.55** | **81.74** |

Table 3: MultiWOZ 2.0 per-domain few-shot joint goal accuracy (JGA). [†]Results from Lin et al. (2021b). *Results from Shin et al. (2022).

| Model | Attraction | | | Hotel | | | Restaurant | | | Taxi | | | Train | | |
|---|---|---|---|---|---|---|---|---|---|---|---|---|---|---|---|
| | 1% | 5% | 10% | 1% | 5% | 10% | 1% | 5% | 10% | 1% | 5% | 10% | 1% | 5% | 10% |
| TransferQA* | 50.25 | 60.92 | 64.28 | 32.46 | 39.02 | 41.99 | 47.12 | 59.16 | 62.24 | 71.12 | 74.47 | 76.07 | 69.01 | 73.17 | 75.46 |
| DS2 - T5* | 60.04 | 68.74 | 70.31 | 43.02 | 48.44 | 50.35 | 56.54 | 65.11 | 67.26 | 76.41 | 79.81 | 80.62 | 73.07 | 76.18 | 77.14 |
| DiSTRICT | **70.71** | **74.98** | **75.30** | **47.35** | **55.37** | **57.78** | **61.03** | **68.65** | **70.41** | **84.06** | **86.06** | **87.68** | **80.31** | **81.41** | **81.91** |

Table 4: MultiWOZ 2.1 per-domain few-shot joint goal accuracy (JGA). *Results from Shin et al. (2022).

over the next best approach on both datasets (i.e, T5DST in MultiWoz 2.0 and TransferQA in Multi-Woz 2.1), and obtains improvements up to 23% on the 'Train' domain.

Both TransferQA and T5DST use hand-crafted prompts, where the former annotates all slots in the form of questions, and the latter uses individual slot descriptions. In the zero-shot setting, this implies that the query-slot is not truly *"unseen"*, since semantic information about the slot is being provided to the model in the hand-crafted prompt. Furthermore, the addition of new domains and slots would first require crafting new prompts, thereby limiting generalizability.

In contrast, DiSTRICT does not possess any additional information about the unseen query slot and instead relies on identifying other semantically similar slots and dialogues from the data available from other domains to enable model reasoning. The improved performance hence reflects the effectiveness of our retriever driven approach in zero-shot settings, and also demonstrates the generalizability of our solution.

**Per-domain few-shot DST**

Tables 3 and 4 show the per-domain few-shot performance on MultiWOZ 2.0 and MultiWOZ 2.1 respectively. DiSTRICT outperforms the baseline approaches across across all domains and across both datasets. For MultiWOZ 2.1, DiSTRICT achieves a JGA improvement of over 15% in the best-case ("Attraction" domain at 1%) and 9% on average, compared to the next best approach, DS2-T5.

Additionally, the availability of even a few labeled examples significantly improves the performance of our retriever, as evidenced by a 46% improvement in JGA on average across all domains over the zero-shot setting from Table 2 with just 1% of available few-shot data in MultiWOZ 2.1, compared to a 34% improvement by TransferQA. This improvement stems from the increased relevance of available in-context examples, since the retriever now has access to a few (i.e. 1%-5%-10%) dialogues from the few-shot domain.

**Cross-domain few-shot and full-shot DST**

From Table 5, we see that DiSTRICT achieves the best performance in the full-shot (100%) setting, obtaining ~ 10% improvement in JGA on average over the other approaches. However, we observe a significant drop in performance in the cross-domain few-shot setting, when the total available training data is reduced. DiSTRICT suffers from a 75% drop in JGA when only 1% of training data is available in MultiWOZ 2.1, compared to a 35% drop for DS2-T5 which suffered the smallest drop in performance.

This performance drop can be attributed to the limited diversity of in-context examples arising from the unavailability of training data. For instance, the 1% setting in MultiWOZ 2.1 translates to the availability of only 84 training examples. Hence, the retriever is limited to repeatedly using the same examples, restricting the model's reasoning capabilities. In contrast, the hand-crafted prompts used by the baseline approaches appear

| Model | MultiWoz 2.0 | | | | MultiWoz 2.1 | | | |
|---|---|---|---|---|---|---|---|---|
| | 1% | 5% | 10% | 100% | 1% | 5% | 10% | 100% |
| TRADE[*] | 11.74 | 32.41 | 37.42 | 48.62 | 12.58 | 31.17 | 36.18 | 46.00 |
| DS2 - BART[*] | - | - | - | - | 28.25 | 37.71 | 40.29 | 46.86 |
| DS2 - T5[*] | **36.15** | **45.14** | 47.61 | 54.78 | 33.76 | 44.20 | 45.38 | 52.32 |
| T5DST[†] | 28.87 | 42.03 | 46.49 | 53.42 | 28.23 | 44.41 | 47.12 | 52.21 |
| IC-DST Codex[§] | - | - | - | - | **43.13** | **47.08** | 48.67 | 50.65 |
| DiSTRICT | 17.13 | 41.88 | **50.39** | **57.02** | 13.39 | 41.31 | **49.73** | **56.08** |

Table 5: Cross domain few-shot and full-shot joint goal accuracy (JGA) on MultiWOZ 2.0 and 2.1. [*]Results from Shin et al. (2022). [†]Results from local implementation based on Lin et al. (2021b). [§]Results from Hu et al. (2022).

to provide sufficient additional information to the model, reducing performance drop in limited data settings.

### 4.6 Additional Experiments

**Impact of number of in-context examples**

We study the effect of varying the retrieved number of in-context examples used in the prompt on DiSTRICT's performance in all our experimental settings. From Figure 3, we observe that using no examples (i.e.) model fine-tuning and inference using only input dialogues, results in very poor performance that is worse than all the baseline comparison approaches. This shows that the addition of relevant examples has a significant impact on conditioning the model for dialogue state tracking.

We also observe an improvement in performance as the number of in-context examples increases, highlighting the potential of using a larger number of examples as part of future work. However, this involves a trade-off, since the improvement is not linear and has diminishing returns, and using a larger number of examples would require more memory, compute resources, and increased training time.

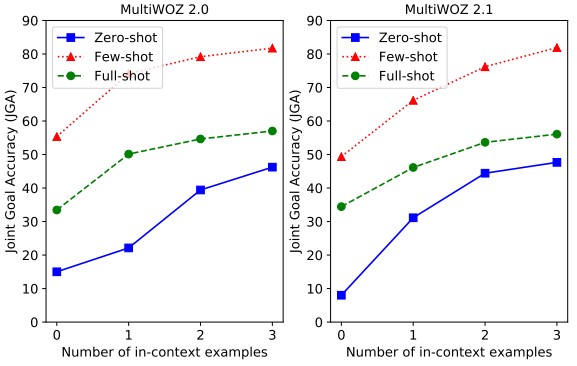

Figure 3: Impact of the number of in-context training examples on joint goal accuracy (JGA) for Zero-shot, Few-shot, and Full-shot settings using the Train domain

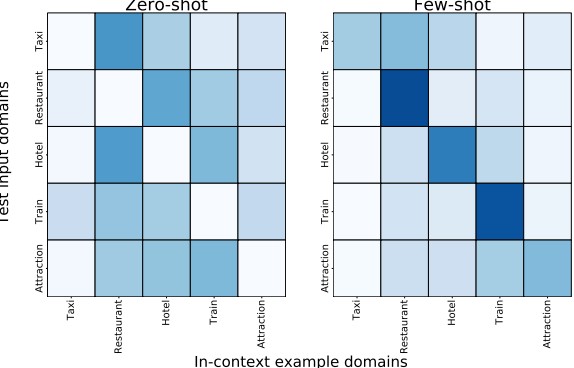

Figure 4: Distribution of in-context examples picked across domains by the retriever in zero-shot and per-domain few-shot (10% data) settings for MultiWOZ 2.1. Darker color reflects a larger number of examples.

**Retriever design**

As described in Section 3.2, DiSTRICT uses the entire dialog context until the current turn as the query to the retriever to identify relevant examples. We compare the performance of this design choice against using only the utterances of the current turn to obtain examples.

Table 6 shows that using the entire context performs better than the single-turn approach for various settings in the MultiWOZ 2.1 dataset. We observed several examples, where the system's response in the current turn was incorrect (e.g. recommending *italian* restaurants instead of *indian* as asked by the user in prior turns). Hence, using only the current turn resulted in low-relevancy examples being retrieved (involving *italian* restaurants), whereas using the entire dialog context ensured the retrieval of more relevant examples (involving *indian* restaurants), thereby demonstrating the importance of using the dialog context.

**Retriever performance**

We examined the effectiveness of the retriever by analyzing the domains and slots that were selected

| Retriever Query | Full-shot | Few-shot Restaurant | Zero-shot Restaurant | Zero-shot Train |
|---|---|---|---|---|
| Single-turn | 52.13 | 67.11 | 22.93 | 45.29 |
| Whole context | 56.08 | 70.07 | 24.01 | 47.66 |

Table 6: Performance comparison using single-turn utterances to retrieve examples vs. using the whole context.

| Case 1 | Slot similarity |
|---|---|
| **Input dialogue** | [System] is there anything else that i can do for you ?
[User] can you find me an expensive place to stay , it should be only 4 stars and include free wifi. |
| **Query slot** + *gold label* | hotel-price range : *expensive* |
| **Retrieved example** | [System] there are several restaurants . what type of food would you like ?
[User] i want somewhere cheap in the centre please |
| **Example slot** | restaurant-price range : cheap |

| Case 2 | Understanding dialogue context |
|---|---|
| **Input dialogue** | [System] is there anything else i can help you with ?
[User] thank you ! i also need to get a taxi to get me to the restaurant by 17:30 . |
| **Query slot** + *gold label* | taxi-arrive by : *17:30* |
| **Retrieved example** | [System] would you like a phone number as well ?
[User] not today , i just need to get a train that will arrive by 08:45 in cambridge |
| **Example slot** | train-arrive by : 08:45 |

| Case 3 | Unrelated slots |
|---|---|
| **Input dialogue** | [System] i am happy to help you find something . do you have a certain area of town in mind ?
[User] centre , please . i want a type of hotel and free parking and free wifi , please . |
| **Query slot** + *gold label* | hotel-internet : *yes* |
| **Retrieved example** | [System] it is a cheap restaurant located in the centre . can i book for you ?
[User] ok book it for 6 on sunday at 15:30 and i need a reference # too |
| **Example slot** | restaurant-area : centre |

Table 7: In-context examples selected by the retriever for different inputs in the zero-shot setting.

for the input dialogues in the test set. Figure 4 shows the heatmaps for zero-shot and per-domain few-shot settings, depicting the relative number of examples picked from each domain for test inputs across all domains.

In the zero-shot setting, since data from the unseen test domain is unavailable, the main diagonal is empty and we observed that examples were relatively evenly picked across the other available domains. In particular, as illustrated by the examples in Table 7, we found that the retriever identified examples containing semantically similar slots or having similar dialogue contexts, thereby demonstrating the effectiveness of our approach.

In the few-shot setting, we observed that the majority of examples were selected from the same domain as the input (i.e. darker diagonals), reflecting the higher semantic and contextual match between intra-domain dialogues. We also studied the distribution of examples at an individual slot level, shown in the appendix, and observed the same patterns. In particular, for the few-shot setting, the retriever prioritized examples containing the same slot, followed by the same domain, validating the

use of semantic matching.

## Impact of model size

| Model | T5-small (60M) | T5-base (220M) | T5-large (770M) |
|---|---|---|---|
| T5DST* | 65.09 | 66.00 | 68.78 |
| DiSTRICT | 66.58 | 67.23 | 70.09 |

Table 8: Impact of model size on JGA for the 'Taxi' domain zero-shot setting in MultiWOZ 2.0. ∗ T5DST results for larger models obtained from a local implementation based on Lin et al. (2021b).

Finally, we studied the effectiveness of using larger models for DST. We evaluate the performance of DiSTRICT and T5DST, as both approaches use the T5-small model (Table 1), with multiple sizes of T5 in the zero-shot setting with the 'Taxi' domain. As shown in Table 8, in both approaches the larger T5-base and T5-large models achieve modest improvements over the T5-small model. However, these improvements may be too limited to justify the potentially significant increase in compute resources required to support larger model sizes in real-world deployments.

| Model | Attraction | Hotel | Restaurant | Taxi | Train |
|---|---|---|---|---|---|
| DiSTRICT | **33.74±0.80** | **22.29±0.29** | **24.17±0.94** | **66.34±0.79** | **46.93±1.64** |
| Variant w/ no fine-tuning | 3.98±0.21 | 1.72±0.15 | 1.10±0.10 | 4.88±0.38 | 2.69±0.11 |
| Variant w/ random examples | 19.83±0.64 | 9.79±0.66 | 14.49±1.08 | 31.49±1.55 | 19.57±0.21 |
| Variant w/ BM25 retriever | 13.91±0.82 | 11.16±0.27 | 11.54±0.77 | 33.87±0.93 | 16.71±0.36 |

Table 9: Ablation study of DiSTRICT using different variants for the zero-shot setting in MultiWOZ 2.1.

**Ablation study**

We compare the performance of DiSTRICT against variants that use the same model, but have differences in the in-context tuning pipeline shown in Figure 2. We evaluate the variants on the zero-shot setting, using the MultiWOZ 2.1 dataset. We define three variants – the first does not perform any fine-tuning (i.e.) in-context learning, using examples from the same retriever as DiSTRICT. The second and third variants perform in-context fine-tuning, but instead use a random retriever (i.e.) select $k$ examples randomly, and a non-parametric BM25 retriever respectively.

The results (Table 9) show that smaller models like 'T5-small' cannot learn to perform DST without any fine-tuning, further highlighting the tradeoffs between performance, model size, and resource requirements. The performance drop with the use of random examples can be attributed to both biasing the model towards incorrect answers corresponding to other domains/slots, and also withholding valuable information that would have been present in the relevant examples used in DiSTRICT. Furthermore, identifying relevant dialogue examples is heavily reliant on their semantic meaning. Hence, functions like BM25 which is based on bag-of-words retrieval, also perform poorly since they ignore semantic similarity and instead rely on word frequency that often does not accurately reflect the meaning of the dialogue. This serves to show that DiSTRICT's retriever driven in-context tuning approach (Figure 2) plays a big role in enabling effective dialogue state tracking.

## 5   Conclusion

We present DiSTRICT, a novel approach for dialogue state tracking using in-context tuning of language models (LMs). For an input dialogue instance and slots, DiSTRICT retrieves the most relevant examples from the training data through semantic matching, and uses these examples as part of the input to the LM to obtain the dialogue state. The fully automated prompt construction, without requiring hand-crafted templates or additional schema information, overcomes drawbacks of prior DST approaches and also reflects the high generalizability of DiSTRICT to new task domains and slots. Our experiments show that DiSTRICT outperforms existing baselines in different zero-shot and few-shot experiments despite using a smaller and lower-resource model. We also demonstrate the effectiveness of our semantic-search based retriever for the DST task and highlight several trade-offs between model performance and resource requirements that impact real-world use. As part of future work, we intend to improve robustness to dialogue quality and distribution-drifts.

## 6   Limitations

The performance of DiSTRICT hinges on the effective retrieval of relevant in-context examples from the training data. This results in our approach being sensitive to issues with data quantity and quality. As shown in our results, when the amount of training data is limited, the retriever often has to select from a pool of examples that have low diversity and semantic similarity to the input, thereby adversely impacting performance.

Additionally, data quality issues such as poorly named slots (i.e. not sufficiently descriptive) and incorrect/mislabeled slot values would also impact the semantic matching and performance of our approach. Also, our zero-shot learning relies on semantic relationships between the unseen samples and the known data. However, if the new task domains are highly disparate from the existing domains, this relationship may not hold, presenting a challenge for zero-shot learning.

Recently, research efforts have studied domain generalization in the context of model robustness under data distribution shifts (i.e.) out-of-distribution (OOD) generalization (Gulrajani and Lopez-Paz, 2020; Venkateswaran et al., 2021a,b, 2023) which can also occur in real-world task oriented dialogue systems. We did not address this as part of our work, and intend to explore OOD model robustness as part of future work.

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

## A Implementation

For the zero-shot and the full-shot experiments, we train our model for 3 epochs and use early stopping based on the loss in the validation set. For the few-shot experiments, we use the models from the zero-shot experiments that were trained on 4 domains (out of 5), and then train them on the fifth domain with 1%, 5%, 10% of the target domain data for 10 epochs. We also use early stopping based on the validation loss.

## B Retriever Performance

We analyzed the distribution of in-context examples at a slot level for different test inputs. Figures 5 and 6 show the heatmap depicting the slots within the in-context examples that were picked for each test query slot for zero-shot and few-shot settings.

For the zero-shot setting, we observed that whenever possible, the retriever prioritized examples containing slots that had a similar semantic meaning as the query slot (e.g.) `restaurant-area` and `attraction-area`, `hotel-price range` and `restaurant-price range`, `train-arrive by` and `taxi-arrive by`. In cases where the query slot had no similar example slots (e.g.) `hotel-internet`, the retriever picked examples based on the dialogue context similarity.

For the few-shot setting, we observed that the retriever prioritized examples containing the same slot as the query, reflected by the dark diagonal in the heatmap. Additionally, the retriever also typically picked examples from the same domain as the test input, which is shown clearly by the clusters within the heatmap. This serves to show that identifying examples using semantic matching is a viable and effective approach.

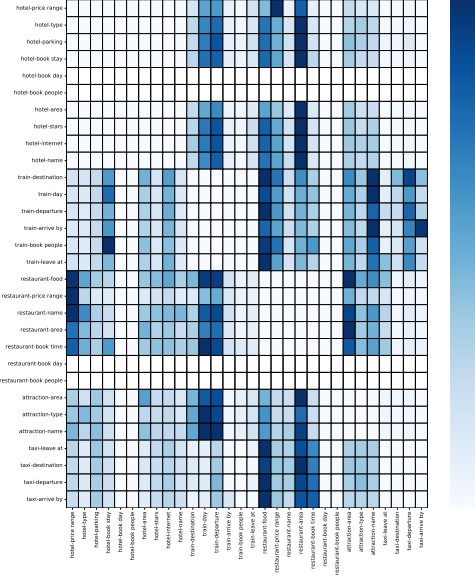

Figure 5: Slot-based heatmap of in-context examples picked by retriever in zero-shot settings

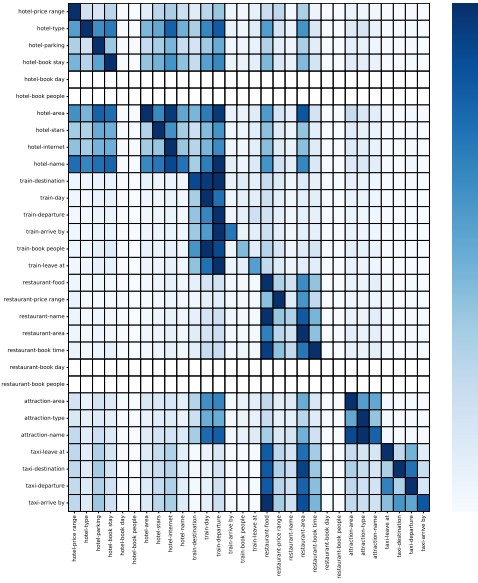

Figure 6: Slot-based heatmap of in-context examples picked by retriever in few-shot settings