# OpenReview forum: "DiSTRICT: Dialogue State Tracking with Retriever Driven In-Context Tuning"
_EMNLP/2023/Conference — EMNLP 2023 Main_

### Official Review · Reviewer_LNYL · 2023-07-27

**Typos Grammar Style And Presentation Improvements:** No obvious issues were found.
**Soundness:** 4

**Excitement:**

3: Ambivalent: It has merits (e.g., it reports state-of-the-art results, the idea is nice), but there are key weaknesses (e.g., it describes incremental work), and it can significantly benefit from another round of revision. However, I won't object to accepting it if my co-reviewers champion it.

**Missing References:**

The discussion on more recent description-drive DST systems is slightly weak.

**Paper Topic And Main Contributions:**

This paper introduces DiSTRICT, a DST system that utilises in-context fine-tuning to achieve better DST performance. The system first retrieves semantically-similar dialogue examples (single turn conversation with ground-truth slot labels) from the training corpus, and then uses them as in-context learning examples during training and testing. The system performs few-shot evaluation by retrieving a small portion of training examples from the domain being evaluated. MultiWOZ 2.0 and 2.1 are used for assessing system performance and several recent pre-trained DST systems are used for comparing zero-shot/few-shot performance.

**Questions For The Authors:**

No particular questions.

**Reasons To Accept:**

1. The writing and experiments are basically clear.
2. The method is straightforward, easy to replicate, and works well. The proposed system outperforms several recent auto-regressive DST systems.
3. The good performance does not rely on exhaustive training data and huge sets of model parameters, which is a good attempt in this domain.

**Reasons To Reject:**

1. The method is not novel and not surprising. In-context learning/tuning is not a novel approach and has been attempted in other work and other domains. As a straightforward approach inspired by recent progress in literature, it achieves good performance without much excitement.
2. The system was evaluated on MultiWOZ only. MultiWOZ is slightly outdated in terms of dataset size, data diversity and label soundness. The system should be evaluated on some more recent DST benchmarks which are more challenging (e.g. Schema-guided DST).
3. The formulation itself has some obvious limitations.
    1. The length of model input can be overwhelming since the input length increases significantly when more examples are used in in-context inference. In the work, K>3 is not attempted (Fig.3) possibly due to this issue;
    2. The system only predicts one slot value at one time, which is a very computationally-expensive setup when a lot of slots need to be classified. Some recent DST systems shifted to predicting multiple slot values at one go (e.g. D3ST) which are far more efficient in practice. The proposed in-context learning approach may work well with this setup (retrieving similar turns with respect to one single slot query), while it may not be efficient enough when predicting multiple slot values given multiple slot queries. More discussion is needed.
4. The statement regarding hand-craft templates is not very convincing. In practice, it is not very difficult to leverage some automated approaches to significantly reduce time and effort in preparing domain/slot descriptions. For example, [1] proposed to leverage paraphrasing to prepare diverse slot descriptions for description-drive DST systems. In practice, it is practically possible to employ several people to work out some slot descriptions for newly-added services in 1 hour. Thus, more discussions are needed to strengthen the advantage of the proposed system relative to those description-drive few-shot/zero-shot DST systems (particularly systems evaluated on the Schema-guided DST dataset).

[1] ****More Robust Schema-Guided Dialogue State Tracking via Tree-Based Paraphrase Ranking**** https://arxiv.org/abs/2303.09905

**Reproducibility:**

4: Could mostly reproduce the results, but there may be some variation because of sample variance or minor variations in their interpretation of the protocol or method.

**Reviewer Confidence:**

4: Quite sure. I tried to check the important points carefully. It's unlikely, though conceivable, that I missed something that should affect my ratings.

---

> ### Author Rebuttal · Authors · 2023-08-28
>
> We thank the reviewer for their feedback and positive comments, and would like to address some of their concerns -
>
> 3a. We acknowledge the reviewer’s concern regarding prompt size. We believe that our design choice of using single turn dialogues as in-context examples rather than the entire dialogue history helps with this issue. We computed the average number of tokens in the prompt for different values of $K$ below, and we see that even with $K=10$, the prompts are within the context window sizes of most popular models.
>
> Prior work [1][2][3] has shown that the addition of more in-context examples in the prompt does not always help and can even cause model performance to drop from SOTA to near-random.
>
> We believe this explains the diminishing returns observed in Figure 3. Additionally, more examples could lead to over-biasing of certain (slot, value) pairs, and we will investigate the performance of our approach with a larger number of examples and update Figure 3 to include these results.
>
> | **K** | **Avg. Token length** |
> |:-----:|:---------------------:|
> | 3     | 181.19                |
> | 5     | 302.08                |
> | 10    | 608.71                |
>
>
> 3b. This is an interesting point,  and we believe that our approach could also potentially leverage the strategy of fine-tuning the model to predict multiple slots simultaneously. Given that most turns provide information about multiple slots, DiSTRICT’s prompt could be modified to include the ground truth value of all these slots. There could also be an interesting tradeoff between efficiency and example quality, and we will add this discussion to the paper.
>
> 4.In some domains, we agree that people could be employed to develop slot descriptions (e.g. using Mechanical Turk). However, in our experience, more complicated domains like some enterprise use-cases require significant domain knowledge, and incorrect descriptions can lead to poor results which is also supported by some of the results shown in the linked paper.
>
> Additionally, the results would rely on the quality of the paraphraser, which can produce incorrect / invalid semantic paraphrases in certain domains. We have ourselves encountered certain situations with the Pegasus model not realizing that Python refers to a programming language and assuming it referred to the snake. This could require further manual annotations to ensure correctness.
>
> Finally, the use of real user utterances as examples could also improve model robustness with the same concept as the paper, since different users use different writing styles, thereby reducing model sensitivity. We thank the reviewer for pointing us to this paper, and will include a discussion and also look at performance on the SGD dataset.
>
> [1] Reynolds, Laria, and Kyle McDonell. "Prompt programming for large language models: Beyond the few-shot paradigm." Extended Abstracts of the 2021 CHI Conference on Human Factors in Computing Systems. 2021.
>
> [2] Min, Sewon, et al. "Rethinking the Role of Demonstrations: What Makes In-Context Learning Work?." Proceedings of the 2022 Conference on Empirical Methods in Natural Language Processing. 2022.
>
> [3] Zhao, Zihao, et al. "Calibrate before use: Improving few-shot performance of language models." International Conference on Machine Learning. PMLR, 2021.

---

### Official Review · Reviewer_nejq · 2023-08-02

**Soundness:** 4

**Excitement:**

4: Strong: This paper deepens the understanding of some phenomenon or lowers the barriers to an existing research direction.

**Paper Topic And Main Contributions:**

This work tries to address the challenge of training a DST system in low-resource scenarios. It proposes to finetune a language model with retrieved in-context exemplars. The authors find improvements compared to previous zero/few-shot approaches.

The author seems to be the first to apply in-context tuning to DST (quite surprisingly). The system design is of minimal prompt crafting effort and is generalizable to different domains.

**Questions For The Authors:**

Question A:
How is the retrieval unit designed? Are the querying and retrieval examples both the concatenation of a query slot and a pair of utterances?

Question B:
In Table 5, DISTRICT has surprisingly low performance in the 1% setting (13% VS. T5DST 28%). Both DISTRICT and T5DST share a similar decoding strategy which is predicting value for a single slot at each pass. It doesn't seem right to have this large gap even if the retrieval pool size is limited (the hypothesis the author makes to explain the drop). Is the problem stemming from the retriever being unable to retrieve relevant examples? 1% is 100 dialogues and breaking into turns you should have 700 turn examples to retrieve, right? If retrieval is essential to the success of ICT then finetuning of the retriever is worth considering.

**Reasons To Accept:**

1. In-context tuning seems to outperform previous systems in various zero and few-shot settings, suggesting the proposed approach can be useful for building real-world DST systems where limited data is available.

2. The paper has compared to appropriate baselines and conducted sufficient experiments to demonstrate the effectiveness.

**Reasons To Reject:**

1. Despite no one has applied in-context tuning to DST before, I personally find the novelty is somehow limited. It's still a new application scenario but it probably falls into the "who does it first" type of paper. I wouldn't position this as a reason to reject, but it's definitely a weakness.

**Reproducibility:**

4: Could mostly reproduce the results, but there may be some variation because of sample variance or minor variations in their interpretation of the protocol or method.

**Reviewer Confidence:**

5: Positive that my evaluation is correct. I read the paper very carefully and I am very familiar with related work.

---

> ### Author Rebuttal · Authors · 2023-08-28
>
> We thank the reviewer for their comments and feedback.
>
> Question A :
>
> The retrieval examples are a concatenation of a single-turn pair of utterances with the slot and its associated ground truth value. The query is the concatenation of the dialogue history till that turn along with the query slot.
> We tried a few different design choices, but this approach gave us the best results.
>
> Question B :
>
> This is a great question! The primary issue we observed was with the **diversity** of dialogues within the 1% setting. With a random selection of dialogues, there were many slots that did not have representative examples within the 1%, and many examples that had the same (slot, value) pairs. This issue, combined with the use of top-k retrieval meant that the model was often shown the **same** set of few-shot examples during fine-tuning. Hence, the model was not learning new information thereby impacting its performance.
>
> On the other hand, since T5DST provides hard-coded information about **all** slots as part of its prompt (irrespective of 1% or 100% dialogues), there were essentially no unseen slots.
>
> This meant that despite using the same decoding strategy, the T5DST model had a lot more information to use during fine-tuning compared to DiSTRICT, and hence its performance did not suffer as much.
>
> We believe that if the 1% dialogues were cherry picked, to be diverse and minimize repetition, the results would be far better. We had not done this, since it would be a biased comparison with prior work.
>
> We also agree with the reviewer’s idea about fine-tuning the retriever, but feel that this might likely require a dataset that has sufficient diversity of slots and domains to ensure the retriever’s effectiveness.

---

### Official Review · Reviewer_sWot · 2023-08-04

**Soundness:** 4

**Excitement:**

3: Ambivalent: It has merits (e.g., it reports state-of-the-art results, the idea is nice), but there are key weaknesses (e.g., it describes incremental work), and it can significantly benefit from another round of revision. However, I won't object to accepting it if my co-reviewers champion it.

**Missing References:**

- [Slot Dependency Modeling for Zero-Shot Cross-Domain Dialogue State Tracking](https://aclanthology.org/2022.coling-1.42) (Wang et al., COLING 2022)
- [Prompter: Zero-shot Adaptive Prefixes for Dialogue State Tracking Domain Adaptation](https://aclanthology.org/2023.acl-long.252) (Aksu et al., ACL 2023)
- [SUMBT: Slot-Utterance Matching for Universal and Scalable Belief Tracking](https://aclanthology.org/P19-1546) (Lee et al., ACL 2019)

**Paper Topic And Main Contributions:**

This research paper investigates a method for in-context fine-tuning for the DST task. The proposed framework, DiSTRICT, begins by retrieving dialogs and slots from historical data that are semantically similar, considering both zero-shot and few-shot scenarios. In the zero-shot setup, similar candidates are selected from source domains, while in the few-shot setup, they are directly chosen from the target domain. These retrieved dialogs are then combined into a prompt without requiring manual template creation. The model is subsequently fine-tuned using these in-context examples and evaluated on test inputs using a similar approach.

**Questions For The Authors:**

Question A. When selecting zero-shot examples from various domains, are you excluding cross-domain dialogs? From Figure 4, I observed that in the zero-shot scenario, the Taxi domain test samples mostly contained examples from the Restaurant and Hotel domains. This occurrence is possibly due to the presence of cross-domain dialogs where users initially book a hotel or a restaurant and subsequently request a taxi. If the model considers these as in-context examples and the gold states are not filtered to exclude taxi slots, the true zero-shot nature might be compromised. However, if the taxi slots are removed from the states, it should still be acceptable even if the model encounters taxi domain discussions in the in-context examples.

Question B. Since you are employing in-context fine-tuning (0-3), I assume the training process is faster. However, why didn't you conduct the experiments multiple times with different seeds? It is essential to observe confidence intervals to ensure the reliability of the results and convince the readers about their trustworthiness.

Question C. The same applies to the ablation part. Why did you only perform it for the train domain? Especially considering that this domain performed the best, it might give the readers the wrong impression that you cherry-picked the results. It would be more comprehensive to conduct the ablation experiments across multiple domains to provide a more balanced and accurate assessment.

**Reasons To Accept:**

- It is the first paper to apply in-context tuning by fine-tuning the LM parameters using the in-context examples retrieved.
- The experiments conducted in the study are thorough and consistently demonstrate incremental enhancements on both Multiwoz 2.0 and 2.1 datasets compared to the baselines.
- The model displays superior performance in both few-shot scenarios and cases with full-shot training.

**Reasons To Reject:**

- The primary hypothesis of the paper, which is in-context fine-tuning, lacks strong support from the experiments conducted. The ablation study demonstrates that fine-tuning improves results only within the 'train' domain. Looking at Table 2, we can observe that the train domain shows the most significant improvement with this method, while other domains exhibit either substantial or no improvements (notably, the differences in JGA between the previous SOTA and DISTRICT model are as follows: Train: +11.1, Attraction: +2.2, Hotel: -0.3, Restaurant: -2.2, Taxi: +4.8). To increase confidence in the findings, it would be more convincing to present how the average results over all domains change in the ablation study. It is unlikely that the authors have chosen to focus on only one domain due to concerns about training time since it should take considerably shorter to do in-context fine-tuning compared to full fine-tuning.
- Moreover, the experiments do not report confidence intervals. This omission raises concerns that the observed improvements might be due to chance rather than an effect of the fine-tuning method.

**Reproducibility:**

5: Could easily reproduce the results.

**Reviewer Confidence:**

5: Positive that my evaluation is correct. I read the paper very carefully and I am very familiar with related work.

**Typos Grammar Style And Presentation Improvements:**

-

---

> ### Author Rebuttal · Authors · 2023-08-28
>
> We thank the reviewer for their feedback and questions, and will include the provided references in the paper.
>
> Question A:
>
> This is a great point, and thank you for pointing this out!
>
> We do indeed filter the gold states to prevent any knowledge leakage of cross-domain slots and maintain the true zero-shot nature. As you have mentioned, we do not filter out the corresponding sentences (like taxi from your example) from the example dialogues.
>
> We will add this clarification to the paper
>
> Question B:
>
> We would like to clarify that with in-context fine-tuning, the model is trained on the exact same number of training examples as with traditional fine-tuning and the difference is the in-context examples that are included with each training example. Hence, the training process takes essentially the same amount of time as traditional fine-tuning (such as those done by prior work).
>
> This is unlike in-context learning which does not involve any additional training at all, but instead just queries the pre-trained model directly. Table 8 shows a performance comparison between in-context fine-tuning and in-context learning.
>
> We acknowledge the reviewer’s comments about confidence intervals, and will be adding them to all our experiments in the paper by re-running with different seeds.
> Below, we updated Table 2 with confidence intervals to demonstrate that the observed performance is not due to chance. Since TRADE and TransferQA do not provide confidence intervals in their published results, we were unable to include those.
>
>
> MultiWOZ 2.0
>
> | **Model** | **Attraction**     | **Hotel**          | **Restaurant**     | **Taxi**           | **Train**          | **Avg.**           |
> |-----------|--------------------|--------------------|--------------------|--------------------|--------------------|--------------------|
> | TRADE     | 19.87              | 13.70              | 11.52              | 60.58              | 22.37              | 25.76              |
> | T5DST     | 33.09$\pm$1.60     | 21.21$\pm$0.61     | **21.82$\pm$0.91** | 65.09$\pm$0.12     | 35.42$\pm$1.42     | 35.20$\pm$0.59     |
> | DiSTRICT  | **33.61$\pm$0.73** | **22.82$\pm$0.43** | 21.30$\pm$1.08     | **66.53$\pm$0.21** | **46.61$\pm$1.56** | **38.17$\pm$0.80** |
>
>
> MultiWOZ 2.1
>
>
> | **Model**  | **Attraction**     | **Hotel**      | **Restaurant** | **Taxi**           | **Train**          | **Avg.**           |
> |------------|--------------------|----------------|----------------|--------------------|--------------------|--------------------|
> | TRADE      | 20.06              | 14.20          | 12.59          | 59.21              | 22.39              | 25.69              |
> | TransferQA | 31.25              | **22.72**      | **26.28**      | 61.87              | 36.72              | 35.77              |
> | DiSTRICT   | **33.74$\pm$0.80** | 22.29$\pm$0.29 | 24.17$\pm$0.94 | **66.34$\pm$0.79** | **46.93$\pm$1.64** | **38.69$\pm$0.89** |
>
>
> Question C:
>
> We acknowledge the reviewer’s feedback and have performed the ablation study on the other domains and present the results below. As stated in the paper, we continue to see that fine-tuning and choice of retriever are both important for DST performance. We will update the paper with these results.
>
> We would also like to address the reviewer’s comment regarding Table 2, and clarify that the comparison approaches (TRADE, T5DST, and TransferQA) all perform fine-tuning with different strategies. Hence, this comparison does not have a direct bearing on our ablation study comparison against a non-fine tuning strategy.
>
> | **Model**                  | **Attraction**     | **Hotel**          | **Restaurant**     | **Taxi**           | **Train**          |
> |----------------------------|--------------------|--------------------|--------------------|--------------------|--------------------|
> | DiSTRICT                   | **33.74$\pm$0.80** | **22.29$\pm$0.29** | **24.17$\pm$0.94** | **66.34$\pm$0.79** | **46.93$\pm$1.64** |
> | Variant w/ no fine-tuning  | 3.98$\pm$0.21      | 1.72$\pm$0.15      | 1.10$\pm$0.10      | 4.88$\pm$0.38      | 2.69$\pm$0.11      |
> | Variant w/ random examples | 19.83$\pm$0.64     | 9.79$\pm$0.66      | 14.49$\pm$1.08     | 31.49$\pm$1.55     | 19.57$\pm$0.21     |
> | Variant w/ BM25 retriever  | 13.91$\pm$0.82     | 11.16$\pm$0.27     | 11.54$\pm$0.77     | 33.87$\pm$0.93     | 16.71$\pm$0.36     |

---

### Meta-Review · Area_Chair_aVGY · 2023-09-19

**Recommendation:** 5

**Metareview:**

The paper presents DiSTRICT, a method to improve DST performance through in-context tuning with relevant retrieved training examples. This is the first work to apply in-context tuning method to the dialogue state tracking task. The selection of baselines are appropriate and the conducted experiments are generally sufficient to demonstrate the effectiveness of the method, while the domain generalizability can be further verified through more detailed analysis, which the authors addressed in the rebuttal.

---

### Decision · Program_Chairs · 2023-10-07

**Decision:**

Accept-Main

**Comment:**

The paper presents DiSTRICT, a method to improve DST performance through in-context tuning with relevant retrieved training examples. This is the first work to apply in-context tuning method to the dialogue state tracking task. The selection of baselines are appropriate and the conducted experiments are generally sufficient to demonstrate the effectiveness of the method, while the domain generalizability can be further verified through more detailed analysis, which the authors addressed in the rebuttal.